behaviour, ecology

extra-pair paternity, mating system, social network analysis, social environment, neighbourhood, *Cyanistes caeruleus*

**Author for correspondence:**
Kristina B. Beck
e-mail: kbeck@orn.mpg.de

# Winter associations predict social and extra-pair mating patterns in a wild songbird

Kristina B. Beck[1], Damien R. Farine[2,3,4] and Bart Kempenaers[1]

[1]Department of Behavioural Ecology and Evolutionary Genetics, Max Planck Institute for Ornithology, Seewiesen, Germany
[2]Department of Collective Behaviour, Max Planck Institute of Animal Behavior, Konstanz, Germany
[3]Department of Biology, University of Konstanz, Germany
[4]Centre for the Advanced Study of Collective Behavior, University of Konstanz, Germany

(iD) KBB, 0000-0002-5027-0207; DRF, 0000-0003-2208-7613; BK, 0000-0002-7505-5458

Despite decades of research, our understanding of the underlying causes of within-population variation in patterns of extra-pair paternity (EPP) remains limited. Previous studies have shown that extra-pair mating decisions are linked to both individual traits and ecological factors. Here, we examine whether social associations among individuals prior to breeding also shape mating patterns, specifically the occurrence of EPP, in a small songbird, the blue tit. We test whether associations during the non-breeding period predict (1) future social pairs, (2) breeding proximity (i.e. the distance between breeding individuals) and (3) the likelihood that individuals have extra-pair young together. Individuals that were more strongly associated (those that foraged more often together) during winter tended to nest closer together. This, by itself, predicts EPP patterns, because most extra-pair sires are close neighbours. However, even after controlling for spatial effects, female–male dyads with stronger social associations prior to breeding were more likely to have extra-pair young. Our findings reveal a carry-over from social associations into future mating decisions. Quantifying the long-term social environment of individuals and studying its dynamics is a promising approach to enhance our understanding of the process of (extra-)pair formation.

## 1. Introduction

Determining the factors that underlie variation in mating behaviour is crucial for our understanding of ecological and evolutionary processes such as sexual selection [1,2], cooperation [3,4] and population demographics [5,6]. In most socially monogamous bird species, some individuals engage in sexual behaviour outside the pair bond resulting in extra-pair paternity (EPP) [7,8]. However, the occurrence and frequency of EPP can vary drastically among individuals, even within the same population [7,9,10]. This variation has previously been linked to differences in individual traits (e.g. male body size [11]; female body condition [12]; male age [13]; sperm morphology [14]; male song characteristics [15]; male plumage [16]; but see [17,18]) or in ecological conditions (e.g. breeding synchrony [19]; breeding density [20]; but see [21,22]). Yet, despite much research, our ability to explain or predict patterns of EPP remains limited.

A major source of variation lies in the social environment. Individuals within a given population do not interact equally with all other members of that population, leading to heterogeneity in the number and quality of social associations. The individual-specific social surrounding should therefore determine important aspects of mating behaviour, such as mate availability, intra-sexual competition and mate choice [23–25]. Individuals usually interact with many more

opposite-sex individuals than expressed in their social pair bond. Thus, the social environment probably includes potential extra-pair mates and may provide the substrate for future extra-pair copulations. For instance, a social surrounding including many opposite-sex members might favour extra-pair behaviour, and the frequency of social associations could be informative about who will mate with whom [23]. However, the effects of the social environment on patterns of EPP have rarely been investigated, despite potentially being able to give valuable insights into the expression of extra-pair behaviour [23,26].

A further limitation to our understanding of mating strategies revolves around the importance and timing of social associations with future (extra-pair) partners, including when decisions about (extra-pair) mating are made. Research on EPP has predominately focused on events or circumstances during the breeding season. For instance, several studies investigated the link between EPP and (a) local breeding density, reflecting the potential number of social associates [20,21,27], (b) the phenotypic composition of the breeding environment [26,28] or (c) associations of opposite-sex individuals during the female's fertile period (e.g. at nest-boxes [29]). However, for many animals, the breeding season is relatively brief and conditions can become suitable for breeding with short notice. By contrast, individuals can interact with others in different contexts for many months prior to breeding. Previous studies suggest that social associations among individuals before the breeding season can translate into the spatial breeding arrangement during spring [30] and potentially predict mating decisions, including social pair formation [31,32] and between-season divorce [33,34]. These findings suggest that social associations prior to breeding may also provide the opportunity for individuals to identify potential extra-pair mates or to form bonds with opposite-sex individuals other than the social mate.

Here, we examine whether social associations prior to the breeding season influence patterns of social and extra-pair mating in blue tits (Cyanistes caeruleus). Blue tits typically form socially monogamous pairs, but frequently engage in extra-pair mating (about half of the broods contain at least one extra-pair young and 10–15% of all offspring are sired by extra-pair males [35,36]). During winter, they forage in flocks including both conspecifics and heterospecifics [37]. Using PIT-tag technology and social network analyses in combination with parentage analysis, we quantified the birds' social associations during foraging events at local bird feeders during winter and monitored their breeding behaviour, including EPP, in the following spring.

We first test whether social associations at bird feeders predict the formation of future social pairs. Second, we examine whether winter associations can predict patterns of EPP. As extra-pair young are usually sired by close neighbours [21], we also test whether social associations at bird feeders during winter predict the observed spatial breeding arrangement—who nests nearby to who—and then examine whether these social associations predict patterns of EPP. Together, these analyses allow us to quantify the likelihood that a female–male dyad will have extra-pair young together while controlling for the two key factors known to influence EPP in blue tits (male age and breeding distance [21]). Our analyses include three variables representing the behaviour during the non-breeding phase: the arrival date of individuals in the local breeding area, the social association strength during foraging and the co-occurrence of individuals at nest-boxes. In blue tits, a

larger difference in arrival date by previous social partners was associated with an increased likelihood of divorce [34]. In the context of this study, we predict (a) that a larger difference in arrival date between two opposite-sex individuals reduces the opportunity to interact and hence leads to a decreased likelihood of having extra-pair young together, and that (b) individuals with stronger social associations during foraging and those that (c) visited a nest-box together during winter will more likely become extra-pair partners. Third, we compare the association strength between social pairs, extra-pair partners and close neighbours. We predict that, if mating outcomes depend on winter social associations, the strength of those associations might be similar for within- and extra-pair partners. By contrast, if extra-pair mating is mainly the result of chance encounters during the fertile period, social pairs will show stronger winter associations than extra-pair partners. Finally, we calculate social networks for each month across the winter to investigate potential temporal patterns of the effects of the winter associations on the likelihood that a female–male dyad will become a social pair or extra-pair partners. Here, we predict that associations closer to the start of breeding are more meaningful in explaining mating patterns and that social pairs show stronger winter associations earlier on compared with extra-pair partners.

## 2. Materials and methods

### (a) Study system

We studied a population of blue tits in a mixed-deciduous oak-dominated forest close to Landsberg am Lech, Germany (Westerholz, 48°08′26″N 10°53′29″E, approx. 40 ha). The study site contains 277 wooden nest-boxes since 2007 and 20 feeders that were put up in the fall of 2017. From November 2017 until mid-March 2018, the feeders provided food (crushed peanuts) ad libitum.

During the 2017 breeding season and the subsequent winter, we trapped blue tits and fitted them with a PIT-tag (transponder), which was implanted under the skin on the back, and a metal ring. We also scored age (yearling versus adult) and took a small (approx. 10 μl) blood sample for parentage analysis and sexing.

All nest-boxes and feeders were equipped with RFID antennas, such that each visit of a PIT-tagged blue tit was automatically recorded [38,39]. For each transponder detection, the bird's identity (unique transponder number), and the date and time were stored on a SD card. From these data, we then extracted information on the co-occurrence of individuals at feeders or nest-boxes and defined the timing of arrival into the study site as the first day an individual was detected (starting on 1 November 2017) either based on PIT tag detection or catching (following [34]). The data relevant for this study were collected between November 2017 and June 2018. For more details on the study system, see [40].

### (b) Foraging associations

The detection of PIT-tagged blue tits at feeders created a temporal data-stream for each location and each day. We used the function 'gmmevents' from the R package 'asnipe' [41] in R (v. 3.5.1 [42]) to assign individuals to temporal clusters reflecting flocking events. This approach uses Gaussian mixture models [32] and generates social association data from sequential detections [43]. We then used the co-occurrence data to calculate the simple ratio index (SRI), defined as: $S_{AB} = x/x + y_{AB} + y_A + y_B$ [44,45]. Here, $S_{AB}$ represents the association strength between individual $A$ and $B$ (i.e. the edge weight in the social network), $x$ is the number of times both individuals co-occurred in the same flock, $y_{AB}$ is the

number of times they were both detected at the same time but not together, $y_A$ is the number of times that $A$ occurred in a flock without $B$ over the time period where both individuals were known to be in the study site, and $y_B$ is the number of times that $B$ occurred in a flock without $A$ over the period where both individuals were known to be in the study site. SRI values can range from 0 (two individuals never associated) to 1 (two individuals were always associated).

We created a non-directional weighted social network from the entire winter period including all individuals and ranked all the associates of a focal individual according to the association index SRI. For instance, if individual A has the following SRI values for three associates: $S_{AB} = 0$, $S_{AC} = 0.5$, $S_{AD} = 1$, the corresponding ranked values would be 1, 2 and 3. We then subtracted 1 from every ranked value and divided the new ranks by the maximum value (2 in our example). This resulted in a 'ranked' association index ranging from 0 (the individual with which the focal individual associated least) to 1 (the individual with which the focal individual associated the most). For each same- and mixed-sex dyad, we then calculated the average of the ranked association index from individual $A$ to individual $B$ and the ranked association index from individual $B$ to individual $A$. From here on, we refer to this average value as the 'winter association strength'.

## (c) Spatial overlap during foraging

We calculated the overlap in spatial activity of each dyad based on the amount of foraging locations that overlapped between the two individuals, as well as the amount of time they spent at these locations, following [30]. This resulted in a value from 0 (no overlap) to 1 (full overlap). For example, when individual $A$ foraged 90% of the time at feeder 1 and 10% at feeder 2, and individual $B$ foraged 90% at feeder 2 and 10% at feeder 1, their overlap in spatial activity would be 0.2 (10% overlap at feeder 1 and 10% overlap at feeder 2).

## (d) Nest-box visits

For each female–male dyad, we quantified co-inspection of nest-boxes during winter (i.e. before the first signs of nest building in the population, which were on 14 March 2018). To find a meaningful definition for the co-occurrence of two individuals at a nest-box, we examined the nest-box visits of future social pair members during winter, because they likely perform nest inspections together. From all recorded visits and for each day and nest-box, we extracted the minimum time difference between the detection of the social female and the detection of her social mate. The minimum time difference between the nest-box visits of two future social partners was on average one minute (s.d. = 16 min, median = 0.02, range: 0–647 min; based on nest-box visits of 101 breeding pairs). Thus, we defined all visits of mixed-sex dyads that occurred within one minute as 'inspecting a nest-box together'. Because the majority of dyads did not visit a nest-box together or only rarely (599 dyads visited a nest-box together, approx. 0.4% of all possible mixed-sex dyads), we defined the co-occurrence at a nest-box as a binary variable (yes/no).

## (e) Spatial breeding arrangement

We examined the spatial breeding arrangement of birds using the R package 'expp' [46]. The package assigns territories to breeding pairs using Thiessen polygons. Based on this information, we determined the neighbourhood order of a focal pair to all breeding pairs in the study site, whereby first-order neighbours refer to neighbours sharing a territory border, second-order neighbours are those that have one territory in between them, and so on (for further details, see [21,46]). To calculate the neighbourhood order, we included all breeding birds, regardless of whether they had been present during winter. We recorded three cases of

social polygyny (a male breeding with two females) during the 2018 breeding season. For these cases, we selected the female with whom the male settled first as 'social female' and assigned the territory accordingly. Further, we excluded five cases of replacement clutches (where either the same pair or the female or male bred again with a different partner after failure of the first clutch).

## (f) Paternity analysis

During the 2018 breeding season, we collected blood samples from all nestlings and breeding adults that had not yet been sampled. We also collected and genotyped all unhatched eggs (provided sufficient DNA could be extracted) and all dead nestlings. Overall, we genotyped 1153 young out of 1238 laid eggs (93%). To assess parentage of all offspring, we used 8–11 microsatellite markers with on average 25 alleles per marker and compared the genotypes of putative parents and all offspring. We then determined how many extra-pair young each male sired with a given female and the number of extra-pair partners for both males and females. For a detailed description of the microsatellite markers and paternity analysis, see [36,40].

## (g) Statistical analyses

For all analyses, we only included data of mixed-sex dyads where both individuals were present during winter (between November 2017 and mid-March 2018) and later bred in the study site. Presence in winter was necessary because some of the null models are based on the behaviour at bird feeders during this period (see below).

Social networks are based on non-independent observations of multiple individuals and thus network measures violate the assumptions underlying most parametric tests [47,48]. We used null models for hypothesis testing to account for non-independence of the data and for non-social factors (e.g. spatial preferences) that may affect the co-occurrence of individuals [47,49]. Specifically, we applied permutation tests by generating replicated datasets from the observed data in which the pattern of interest, in our case the associations among individuals, is randomized [47,49]. We then calculated the same test statistic for 1000 randomly generated datasets as for the observed data. If the value of the test statistic from the observed data fell outside the 95% range of values generated by the permutations, the effect was considered statistically significant. Details of each null model (i.e. for each hypothesis test) are given below.

### (i) Do winter associations predict future social pairs?

We tested whether the winter association strength predicts whether a given female–male combination will become a social (breeding) pair, using a logistic matrix regression [50] with the 'netlogit' function of the R package 'sna' [51]. We included as the dependent variable whether a female–male combination ended up as a social breeding pair (yes/no). The only explanatory variable was the winter association strength of each dyad. We examined the effect of winter association strength by performing 1000 permutations using a customized null model. We randomized the winter association strength across all dyads within the same neighbourhood order and randomized the association strength of social pairs within the first-order neighbours. Thus, we kept the spatial breeding structure of all individuals, but permuted the winter association strength among all opposite-sex conspecifics.

Some birds were only equipped with a transponder for part of the time during the study period. This means that they could have been part of the study population for an unknown period without having been detected, which may lead to a wrong representation of the social associations of these birds with others. Therefore, we repeated the analyses only including birds that had been equipped with a transponder prior to the start of the study.

## (ii) Do winter associations predict spatial breeding arrangement?

We examined whether winter association strength predicts the breeding proximity of birds using a linear matrix regression with the 'netlm' function of the R package 'sna' [51] excluding social pairs from the dataset. We used a model with breeding proximity (i.e. the neighbourhood order, ranging from 1–5) of a dyad as the dependent variable and their winter association strength as well as their overlap in spatial activity during foraging as independent variables. We scaled all independent variables by dividing each value by two times the standard deviation to allow direct comparison of effect sizes among variables [52].

The breeding proximity of two individuals may simply reflect similar spatial preferences, and not the fact that they associated with each other (i.e. foraged together). To determine the effect of winter association strength, we performed 1000 permutations using a spatially restricted node permutation. We disentangled spatial and social effects by randomly reassigning the social network position of each individual to another individual that visited the same feeder before the start of the breeding season (i.e. the last feeder a bird was recorded at). For example, if individuals $A$ and $B$ were both recorded last at feeder 1, the social network positions of individuals $A$ and $B$ would be swapped. If only the preference for the same spatial location determines the associations of $A$ and $B$ with conspecifics, we would expect no difference between the observed and the permuted data. However, if individuals share the same spatial location but differ in their associations with other blue tits, the observed data will differ from the randomized data. Further, we repeated the analyses including only birds that had been equipped with a transponder before the start of the study.

Lastly, we compared the winter association strength between all neighbourhood orders. For each of the five neighbourhood orders, we determined the average winter association strength and calculated the difference in the means between all possible comparisons. We then randomly sampled the winter association strengths among the five orders, and again calculated the difference in the means between all order comparisons. We repeated this 1000 times and compared the differences calculated from the randomized data with the actual difference calculated from the observed data.

## (iii) Do winter associations predict extra-pair paternity?

We examined whether we can predict the likelihood of a female–male combination to be classified as extra-pair partners from the winter association strength by performing a logistic matrix regression [50] using the 'netlogit' function [51]. We used the same dataset as described above (containing mixed-sex dyads and excluding social pairs). Whether a female–male combination had extra-pair young together (yes/no) was the dependent variable. As independent variables, we included breeding distance (i.e. neighbourhood order, ranging from 1 to 5 [21]), male age (yearling versus adult [2]) and three factors describing the behaviour during winter: (1) winter association strength of each dyad, (2) the absolute difference in arrival time between members of each dyad and (3) whether both individuals inspected a nest-box together (yes/no) during winter. We scaled all independent variables by dividing each value by two times the standard deviation [52]. We examined the effect of winter association strength using the same null model described in the section on social pairs.

We repeated the analyses on EPP on a smaller spatial scale, including only direct and second-order neighbours, because the majority of extra-pair sires bred within this neighbourhood (see results). We also repeated the analyses including only birds that had been equipped with a transponder before the start of the study.

Lastly, we examined whether the effect of winter association strength on EPP simply arises from the potential carry-over effects of the previous social breeding structure. We repeated

the analyses with two datasets: (1) using dyads where both partners had bred in the study site in the previous season (2017) and (2) using dyads where at least one individual bred for the first time in the study site, which excludes the possibility of carry-over effects. If significant, this test provides evidence that the effect of winter social associations on patterns of EPP is not simply a by-product of the previous breeding associations.

## (iv) Comparing the association strength between social partners, extra-pair partners and other close neighbours

We examined whether winter association strength differed between social pairs, extra-pair partners, direct neighbours and second-order neighbours. For each of the four categories of relationships, we determined the average winter association strength and calculated the difference in the means between all possible categories (e.g. social pairs, extra-pair partners, etc). Next, we randomly sampled the winter association strengths among the four categories and again calculated the difference in the means between all categories. We repeated 1000 times and inferred statistical significance by comparing the differences calculated from the randomized data to the actual difference calculated from the observed data.

## (v) Temporal changes in the social network

The effect of winter association strength on the likelihood that a female–male dyad ends up as social pair or extra-pair partners may change during the study period. For example, associations closer to the start of breeding might be more meaningful in explaining mating patterns. Furthermore, the strength of the associations with the social (breeding) and the extra-pair partner may change over time. For example, social pairs may show stronger winter associations earlier on compared to extra-pair partners or the relative association strength of within- and extra-pair partners may change as birds anticipate the breeding season. To examine potential temporal differences in the effect of winter association strength, we created the same network as described above, but for each month separately (i.e. one network for November, December, January, February and 1–14 March). We then repeated the analyses to test whether winter association strength predicts whether a given female–male combination will become extra-pair partners or a social pair, as described in the sections (i) and (iii) above.

We also post-hoc split the winter into two halves (calculating one network for early winter: November–January) and one for late winter (February 1–14 March) and repeated all analyses as explained above.

## 3. Results

During the 5-month study, we recorded 30 205 flocking events at feeders (on average 15 per feeder per day including on average 4 individuals per flock, range = 1–42), comprising 563 individuals. Individuals were present on average 46 days (s.d. = 40.4, range: 1–138) and used 7.5 feeder locations (s.d. = 4.5, range: 1–20). From the 563 individuals recorded during winter, 221 (approx. 39%) bred in the subsequent spring. During the breeding season (14 March–25 June), we recorded 124 social pairs (excluding replacement clutches and cases of polygyny; see Materials and methods), i.e. 248 individuals (of which 221 (89%) were present during winter). Approximately 41% of nests contained at least one extra-pair young (range: 1–11 EPY per nest, mean = 2). In total, 59 dyads involving 95 individuals had extra-pair young ($N_{Females} = 49$, $N_{Males} = 46$). Of those 95 individuals, 64 (approx. 67%) were present during winter.

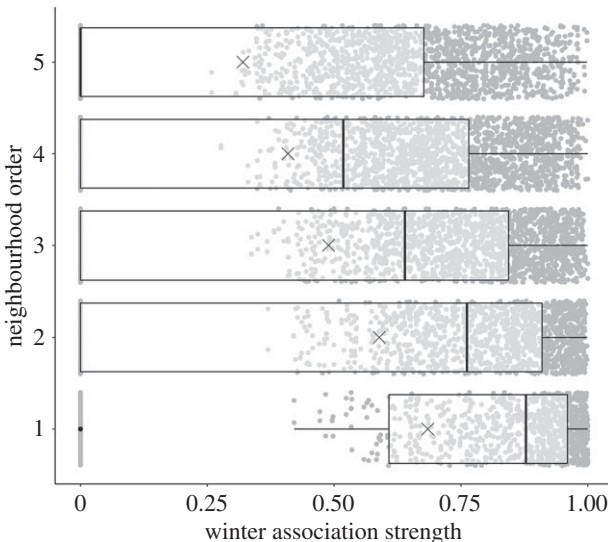

**Figure 1.** Relationship between breeding distance (neighbourhood order: 1 = direct neighbours, 2 = second-order neighbours, etc.) and winter association strength. Boxplots show the minimum values, lower quartile, median, upper quartile, maximum values and outliers (indicated as black dots). The mean is indicated by a cross. Grey points show all data. The mean winter association strength differed significantly between all neighbourhood order comparisons.

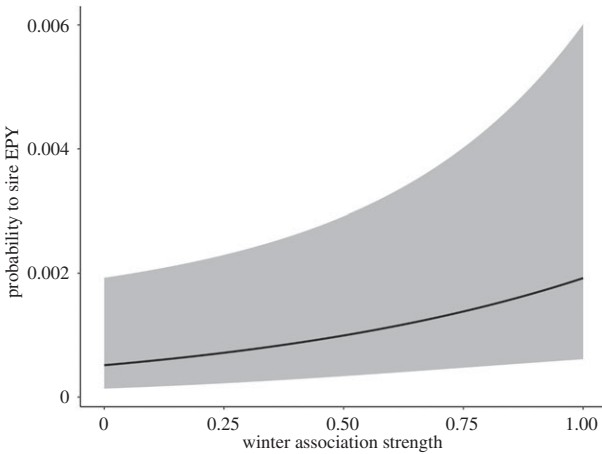

**Figure 2.** The predicted probability for a female–male dyad to have EPY together in the subsequent breeding season in relation to its winter association strength (while keeping all other independent variables constant at their mean values). The grey ribbon shows the 95% confidence interval from a generalized linear-mixed model including neighbourhood order, male age, box visit and difference in arrival as independent variables and including male and female identity as random effects.

### (a) Winter associations predict future social pairs

The association strength during winter was a significant predictor of whether a female–male dyad ended up as a social breeding pair ($N_{Dyads} = 12168$, $N_{Females} = 117$, $N_{Males} = 104$; estimate: 5.23, permutation test: $p < 0.001$). The results did not change qualitatively when the analysis only included individuals that had been equipped with a transponder before the start of the study ($N_{Dyads} = 7622$, $N_{Females} = 74$, $N_{Males} = 103$; estimate: 6.16, permutation test: $p < 0.001$).

### (b) Winter associations predict spatial breeding arrangement

Individuals with a larger overlap in spatial activity during winter (estimate: $-0.96$, $p < 0.001$) and those with stronger winter associations during foraging (estimate: $-0.17$, permutation test: $p < 0.001$) ended up breeding closer together (figure 1). The results did not change qualitatively when the analysis only included individuals equipped with a transponder before the start of the study (electronic supplementary material, table S1). The mean winter association strength differed significantly between all neighbourhood orders (mean winter association strength: first order: $0.69 \pm 0.38$ s.d., second order: $0.59 \pm 0.59$ s.d., third order: $0.49 \pm 0.39$ s.d., fourth order: $0.41 \pm 0.38$ s.d., fifth order: $0.32 \pm 0.36$ s.d.; $p < 0.001$ for all comparisons; figure 1).

### (c) Winter associations predict extra-pair paternity

Female–male dyads that were more strongly associated in winter were more likely to have extra-pair young together (mean winter association strength $\pm$ s.d.; EP partners: $0.75 \pm 0.31$, remaining neighbours: $0.32 \pm 0.37$; figure 2, table 1), independently of the spatial component (see corresponding null model). Further, female–male dyads that had visited a nest-box together before the breeding season were more likely to become extra-pair partners (percentage of pairs that visited a nest-box: EP partners: 23%, remaining neighbours:

2%; table 1), whereas the difference in arrival date did not have an effect (table 1).

The majority of nest-box visits were performed in late winter (January–mid-March). During this period, on average 13 unique dyads visited a nest-box on a given day (range: 1–37). During early winter (November–December), on average only 2 dyads visited a box on a given day (range: 1–4; electronic supplementary material, figure S1). Those birds that inspected a box did it on average with 2.3 other individuals (range: 1–8; excluding the future social partner). The number of partners with whom a bird visited a nest-box did not differ between faithful and unfaithful individuals (unfaithful: mean = 2.1, range: 1–7, faithful: 2.4, 1–8; Wilcoxon rank sum test: $W = 1288$, $p = 0.56$).

As in previous studies on blue tits, older males were more likely to sire extra-pair young (table 1) and the majority of extra-pair sires bred within the close neighbourhood (51% and 32% of extra-pair sires were first- and second-order neighbours, respectively). When the analysis was restricted to first- and second-order neighbours, the effect of winter association strength was similar in size, but no longer significant (table 1). The results did not change qualitatively when only individuals which had been equipped with a transponder before the start of the study were included (electronic supplementary material, table S2) or when running the analysis separately for dyads where both partners had bred in our study site in 2017 and for dyads where at least one bird was unfamiliar to the site. For both datasets, individuals with a higher association strength were more likely to have extra-pair young together (electronic supplementary material, table S3).

### (d) Comparing the association strength between social partners, extra-pair partners and other close neighbours

Association strength was highest for social pairs and lowest for second-order neighbours (figure 3). Winter association strength did not differ significantly between social pairs (mean $\pm$ s.d.: $0.83 \pm 0.30$) and extra-pair partners ($0.75 \pm 0.31$, permutation

**Table 1.** Results of logistic network regression models examining the effect of winter association strength on the likelihood of a female–male dyad to have extra-pair young together. The first model included all neighbourhoods (first to fifth order). The second model included only first- and second-order neighbourhoods. p-values inferred from the permutation tests are shown in italic.

| | all neighbourhoods | | | first- and second-order neighbourhood | | |
|---|---|---|---|---|---|---|
| | estimate | exp(b) | *p* | estimate | exp(b) | *p* |
| intercept | −6.34 | 0.002 | | −4.35 | 0.01 | |
| neighbourhood order | −2.48 | 0.08 | <0.001 | −0.90 | 0.41 | 0.02 |
| male age[a] | 0.69 | 2.00 | 0.04 | 0.99 | 2.70 | 0.01 |
| winter association strength | 0.97 | 2.63 | *0.01* | 0.72 | 2.05 | *0.06* |
| box visit[b] | 0.40 | 1.50 | 0.01 | 0.66 | 1.93 | 0.01 |
| difference in arrival time | 0.53 | 1.70 | 0.16 | 0.19 | 1.21 | 0.65 |

[a]Adults compared with yearlings.
[b]Visiting a box together before the start of breeding (compared with no visit).

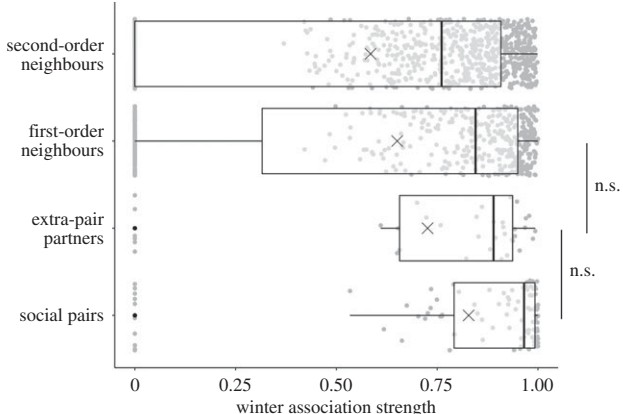

**Figure 3.** The winter association strength for different categories of female–male pairs. Boxplots show the minimum values, lower quartile, median, upper quartile, maximum values and outliers (indicated as black dots). The mean is indicated by a cross. Grey points show all data. Horizontal lines connect pair categories that do not differ significantly. Sample sizes for the different female–male dyads: $N_{Social\ pairs} = 99$, $N_{Extra-pair\ partners} = 37$, $N_{first-order\ neighbours} = 500$, $N_{second-order\ neighbours} = 937$.

test: $p = 0.18$; figure 3; electronic supplementary material, figure S2). However, the association strength also did not differ between extra-pair partners and direct neighbours ($0.65 \pm 0.39$, permutation test: $p = 0.28$). Social pairs had significantly stronger associations compared to direct or second-order neighbours ($0.58 \pm 0.39$, permutation test: both $p < 0.001$).

### (e) Temporal changes in the social network

In general, the effect sizes of the association strength as predictor of mating increased as the breeding season approached (electronic supplementary material, figure S3). The strength of association in late winter (March) significantly predicted which female–male dyad ended up as extra-pair partners ($p < 0.001$), while the association strength earlier in winter (January–March) significantly predicted the likelihood of a dyad ending up as a social breeding pair (January: $p = 0.004$, February: $p = 0.007$, March: $p < 0.001$).

Effect sizes and *p*-values for analyses on social networks generated for the early (November–January) and late winter period (February–March) can be found in the electronic supplementary material (figure S4).

## 4. Discussion

It has been suggested that EPP emerges from the social interactions among multiple individuals (i.e. the focal male or female, their social partner and the potential extra-pair mate(s) [9,23]). Here, we provide extensive empirical support for this idea. We show that social associations during winter carry-over into the spatial breeding arrangement, whereby stronger associated individuals subsequently nested more closely together. This, by itself, will make it more likely that they end up becoming extra-pair mates, because extra-pair sires are typically close neighbours. However, independently of this spatial component, our results show that female–male dyads with stronger associations during winter are more likely to have extra-pair young together. Our study thus suggests that associations prior to breeding influence future mating behaviour.

The maintenance of social bonds with conspecifics can provide several benefits [53] such as reduced aggression [54], better access to information [55,56], increased opportunities for cooperation [57,58] or increased survival [59,60]. Thus, during a prolonged stationary period such as breeding, individuals might benefit from positioning themselves in a suitable social environment. Here, we show that associations during foraging prior to breeding carried over into the spatial breeding arrangement of blue tits (figure 1), similar to what has been found in the closely related great tit [30]. In great tits, familiarity with breeding neighbours increased reproductive success [61]. In cowbirds (*Molothrus ater*), females who spent more time with familiar individuals during the non-breeding phase laid more eggs in the subsequent breeding season [62]. Although the mechanisms underlying such effects are not yet clear, the potential benefits gained from having a familiar social surrounding during breeding may cause individuals to preferably nest closer to conspecifics they are more strongly associated with.

Familiarity to breeding neighbours may also facilitate opportunities for extra-pair matings. In many species, including blue tits, extra-pair young are mostly sired by neighbouring males (e.g. [29,63]). This raises the question whether EPP is simply the result of coincidental meetings between close neighbours or whether it emerges from social preferences for specific mating partners or is at least facilitated by previous social interactions. A previous study found no evidence that the proportion of familiar neighbours (i.e. familiar from

previous breeding seasons) or the presence of a former social mate influenced the patterns of EPP [26]. Here we examined the associations among individuals that arose during the preceding non-breeding season. Our study shows that individuals that were more often foraging together in winter and those that visited a nest-box together were more likely to end up as extra-pair partners (figure 2 and table 1). Previously, Schlicht *et al.* [29] showed that the nest-box visits of males to neighbouring females during their fertile period also predicted the likelihood of having extra-pair young together, and our findings corroborate these results. Even though spatial proximity was the strongest predictor of the occurrence of EPP (the effect size for spatial proximity was more than double that of the winter association strength when considering all neighbourhoods, but effect sizes became more similar when only considering the close neighbourhood; table 1), our findings suggest that EPP does not only arise from spatial proximity and mating opportunities during breeding, but that they are also predicted by prior associations, especially those that took place closer to the start of the breeding season (electronic supplementary material, figure S3).

Winter social associations may simply reflect the preceding breeding social structure. However, when repeating the analyses only including dyads where at least one individual was not present, during the previous breeding season, the winter association strength still predicted extra-pair mating patterns. This indicates that winter social associations and their effect on EPP cannot solely be a consequence of the social structure during the previous breeding season. This finding makes logical sense, especially for short-lived species like the blue tit, as all individuals will necessarily experience a winter flocking period prior to first reproduction, and many individuals will reproduce only once in their life. However, our findings raise questions about whether the increased probability to have EPP with familiar individuals is due to mating preferences taking place prior to breeding (i.e. social associations are driven by extra-pair mate choice) or whether extra-pair matings are indirectly facilitated by other social processes (e.g. reduced aggression due to familiarity).

During winter, we found a clear negative gradient in the association strengths across what could be predicted as a spectrum of future reproductive engagement. Future social partners had the strongest association through to future second-order neighbours having the weakest. Importantly, the differences in winter association strength between future social pairs, extra-pair mates and close neighbours were small (figure 3), highlighting the substantial potential for reproductive outcomes to be shaped by over-winter associations. It could be that these results are explained by methodological limitations of our study. We measured associations exclusively based on foraging events and blue tits usually forage in flocks. Therefore, future social partners and neighbours (including future extra-pair partners) probably foraged together many times and hence may end up having similar association strengths.

Information about fine-scale associations within flocks would help to conclusively show that social interactions with future extra-pair mates differ from other close neighbours. To fully understand whether and how prior social associations affect mating patterns, studies using more advanced tracking technologies [64] are needed to capture finer-scale patterns of social preferences. Furthermore, studies examining how differences in winter social structure (e.g. populations with varying levels of fission-fusion dynamics or with varying turn-over rates) affect future mating decisions would improve our understanding of mating patterns.

When and how individuals make mating decisions is still largely unknown. We assessed whether the importance of winter social associations as a predictor of future mating patterns changes over the season. Perhaps unsurprisingly, we find that the effects of social associations increased both for social pairs and extra-pair partners as the breeding season approached (electronic supplementary material, figure S3). For extra-pair partners, the effect of association was strongest in late winter, whereas the effect on social pairs was clear throughout the winter. While this pattern could simply be caused by lower statistical power at the beginning of the study (i.e. less individuals were present in November than in March), the conclusions seem robust. When we split the study period in early and a late winter, association strength in both periods significantly predicted future social pairs, whereas only the association strength in late winter predicted whether two individuals became extra-pair partners (electronic supplementary material, figure S4). These findings suggest that social pair bonds are established earlier than associations with extra-pair partners, thus providing new insights into the dynamics of different types of social relationships.

**Ethics.** Permits were obtained from the Bavarian government and the Bavarian regional office for forestry (LWF).

**Data accessibility.** Data available from the Dryad Digital Repository: https://doi.org/10.5061/dryad.rv15dv44s [65].

**Authors' contributions.** All authors conceived the idea and designed the study; B.K. conducted the paternity analyses; K.B.B. and D.R.F. analysed the data with input from B.K.; K.B.B., D.R.F. and B.K. wrote the manuscript.

**Competing interests.** We declare we have no competing interests.

**Funding.** This work was supported by the Max Planck Society. D.R.F. received additional funding from the Deutsche Forschungsgemeinschaft (DFG grant FA 1402/4-1) and the DFG Centre of Excellence 2117 'Centre for the Advanced Study of Collective Behaviour' (ID: 422037984). K.B.B. is a PhD student in the International Max Planck Research School for Organismal Biology.

**Acknowledgments.** We are grateful to all people who contributed to data collection, in particular Agnes Türk, Andrea Wittenzellner, Carles Durà, Cécile Vansteenberghe, Friederike Böhm and Giulia Bambini. We thank Sylvia Kuhn and Alexander Girg for microsatellite genotyping; Peter Loës and Peter Skripsky for feeder and nest-box development and maintenance; Mihai Valcu for helpful comments on the analyses; and Daniel Costa and two anonymous reviewers for constructive comments on the manuscript. We thank the Bavarian regional office for forestry (LWF) for permission to work in Westerholz.

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
