## [Reviewer comments · Proceedings of the Royal Society B: Biological Sciences]

Review History

RSPB-2019-2606.R0 (Original submission)

Review form: Reviewer 1

Recommendation

Accept with minor revision (please list in comments)

Scientific importance: Is the manuscript an original and important contribution to its field?

Excellent

General interest: Is the paper of sufficient general interest?

Excellent

Quality of the paper: Is the overall quality of the paper suitable?

Excellent

Is the length of the paper justified?

Yes

Should the paper be seen by a specialist statistical reviewer?

No

Do you have any concerns about statistical analyses in this paper? If so, please specify them explicitly in your report.

No

It is a condition of publication that authors make their supporting data, code and materials available - either as supplementary material or hosted in an external repository. Please rate, if applicable, the supporting data on the following criteria.

Is it accessible?

Yes

Is it clear?

Yes

Is it adequate?

Yes

Do you have any ethical concerns with this paper?

No

Comments to the Author

This is an excellent study that provides probably the most detailed analysis of spatial and social relationships between breeding birds within a population, ever undertaken. This framework, achieved through an intensively tagged and monitored (with pit tags and decoders on feeders and nest boxes), provides an excellent opportunity to then investigate the effects of social and spatial structuring on patterns of infidelity in this socially monogamous bird. The question is an interesting one, given that despite many decades of study we still don't fully understand the variation in extrapair paternity across the individuals in a population.

The introduction and methods are both very well written and I have no general or specific comments on those sections.

Whilst there is no problem with the data and results presented in the study, with respect to the winter associations between male-female dyads and both social pairs and extrapair relationships it would be really good if the authors are able to provide a bit more information to help characterise these dyadic interactions. For example, the data is really interesting on male-female dyadic visits to nest boxes, but it would be useful to know a little more about this behaviour. For example for each focal male (or female), on average, with how many other individuals are such dyadic visits to nest boxes made? Does that differ between individuals that are subsequently faithful and unfaithful individuals? Are such dyadic visits made by individuals that are already in pairs? i.e. if a pair were in a social partnership the previous spring with the partnership persisting into the subsequent year.

Finally, it would also be useful if the authors are able to characterise the temporal patterns with respect to the dyadic nest box visits. Were they mostly in the late winter period closest to nesting, or did they occur in the early winter period.

I believe that the additional focus on the nest box visits is useful because this behaviour does appear to be more likely related to mate choice and pair settlement. The other form of data (social associations at feeders), is more likely to simply reflect the spatial distribution of birds and more incidental associations between neighbouring birds (although I take the point that this has been accounted for in the analyses presented).

Is it possible for the authors to make it a little clearer, the extent to which their main results can be driven by a) the spatial proximity of individuals, and the social associations. It seems clear enough that both are contributing to the incidence of extrapair paternity, but it would be useful if

they are able to describe that clearly in percentage terms. Would be useful to make a clear statement about this in the discussion.

I was somewhat surprised at the relatively high number of individuals that were present in the winter, but not in the subsequent spring. Perhaps that is because in Europe, as opposed to populations in the UK, there is a lot of mobility across populations. Would be useful of the authors can discuss this a little more. For example is it the case that a lot of these birds have come from further north to over-winter in this area? Furthermore, with respect to the main findings, what would we expect in other populations such as those studied in the UK, where a higher proportion of the birds that are found in the winter will be present in the spring?

Review form: Reviewer 2

Recommendation

Accept with minor revision (please list in comments)

Scientific importance: Is the manuscript an original and important contribution to its field?

Excellent

General interest: Is the paper of sufficient general interest?

Excellent

Quality of the paper: Is the overall quality of the paper suitable?

Excellent

Is the length of the paper justified?

Yes

Should the paper be seen by a specialist statistical reviewer?

No

Do you have any concerns about statistical analyses in this paper? If so, please specify them explicitly in your report.

No

It is a condition of publication that authors make their supporting data, code and materials available - either as supplementary material or hosted in an external repository. Please rate, if applicable, the supporting data on the following criteria.

Is it accessible?

N/A

Is it clear?

N/A

Is it adequate?

N/A

Do you have any ethical concerns with this paper?

No

Comments to the Author

This observational study uses technology in combination with social network analysis as applied to an exceptional (possibly unique) data set to better understand the proximate determinants of social pair formation and, in particular, between-individual variation in extra-pair paternity (EPP) in the socially monogamous blue tit. This excellent study addresses a generally very important topic which will be of major interest to readers of Proceedings B. The results are intriguing, the text is written exceptionally well and the authors do a particularly good job in communicating a relatively complex pipeline of statistical analyses and the corresponding results in a very concise manner. I have only one major point which the authors should consider discussing more (see below) and in addition a number of more specific comments which mainly point to opportunities for improving the presentation.

Major comments

To what degree do “winter associations” (as the main predictor of interest in this contribution) themselves represent carry-over effects from the preceding breeding season? For example, for non-yearling birds (as of start of the study in November), associations in winter may simply reflect social network structure from last spring’s breeding neighbourhood. In this context, the authors refer to an as yet unpublished study (reference 22) addressing effects of “changes in neighbourhood” (between breeding seasons) on within-individual variation in extra-pair paternity. Lacking access to this piece of work, I am not sure whether the documented “little effect” can rule out a correlation/carry-over between associations during the preceding breeding season and associations in winter. Experimental manipulation of winter associations is probably very difficult if not impossible, but carry-over effects from the preceding breeding neighbourhood should not apply to yearling birds (as of start of the study in November). It could thus be interesting to explore whether effects of winter association on subsequent mating patterns differ by age category by including interaction terms (age category/dyad age combination-by-winter association). For example, if the reported significant main effect of winter association on EPP is mainly driven by dyads involving non-yearling birds (as of start of the study in November), this would be suggestive of carry-over effects from the preceding breeding neighbourhood to winter time.

Specific comments (lines)

- 54 and 91: socially monogamous OR socially-monogamous (but ideally consistently).
- The correct statement in lines 60-61 would be supported more convincingly if also null results were featured in the preceding sentence (lines 57 to 60).
- 167: Delete redundant “initial”.
- 209: Please add some information on the observed variation in minimum time differences here (e.g. SD and range).
- 292: Delete redundant “in winter”.
- 327: Please use wording that more clearly indicates whether 89% refers to pairs or individuals.
- 334: Give N.
- 471/ Figure 1: For the analysis visualised in Figure 1, neighbourhood order is the dependent and winter association strength is the independent variable (as also explicitly stated in line 476 nearby). Therefore I do not understand why neighbourhood order is on the x and winter association strength is on the y in Figure 1?
- 482-483/ legend of Figure 2: Figure 2 visualizes model predictions including uncertainty and thus the legend should state this and inform about the model underlying the predictions. For example, are the presented predictions controlled for other significant effects in the corresponding model (e.g. age)?
- 493/ Figure 3: For x axis labeling it seems preferable to use exactly the same terms as established and used in the text, e.g. “2nd order neighbours” instead of “2nd direct neighbours” (also applies to supplement); the abbreviation “EP” can be easily guessed but is nowhere defined.
- 519-522/ Table 1: The text states a positive relationship but both estimates are negative. It will less likely cause confusion for readers trying to understand the table content without reference to

the text, when in line 520 the measure of breeding proximity (neighbourhood order) is given. Also give N.
-537: Replace "direct" with "1st".
-538: Replace "neighbours" with "neighbourhoods".

Decision letter (RSPB-2019-2606.R0)

17-Dec-2019

Dear Miss Beck:

Your manuscript has now been peer reviewed and the reviews have been assessed by an Associate Editor. The reviewers' comments (not including confidential comments to the Editor) and the comments from the Associate Editor are included at the end of this email for your reference. As you will see, the reviewers and the Editors have raised some concerns with your manuscript and we would like to invite you to revise your manuscript to address them.

Research ethics:

Use of animals and field studies:

Please submit a copy of your revised paper within three weeks. If we do not hear from you within this time your manuscript will be rejected. If you are unable to meet this deadline please let us know as soon as possible, as we may be able to grant a short extension.

Best wishes,
Dr Daniel Costa
<mailto:proceedingsb@royalsociety.org>

Associate Editor
Board Member: 1
Comments to Author:

Great paper and a wonderful and important contribution to the field. Is there a reason why male age, but not female age is used in the analysis? Would it be possible to add that?

Reviewer(s)' Comments to Author:

Referee: 1

Comments to the Author(s)

This is an excellent study that provides probably the most detailed analysis of spatial and social relationships between breeding birds within a population, ever undertaken. This framework, achieved through an intensively tagged and monitored (with pit tags and decoders on feeders and nest boxes), provides an excellent opportunity to then investigate the effects of social and spatial structuring on patterns of infidelity in this socially monogamous bird. The question is an interesting one, given that despite many decades of study we still don't fully understand the variation in extrapair paternity across the individuals in a population.

The introduction and methods are both very well written and I have no general or specific comments on those sections.

Whilst there is no problem with the data and results presented in the study, with respect to the winter associations between male-female dyads and both social pairs and extrapair relationships it would be really good if the authors are able to provide a bit more information to help characterise these dyadic interactions. For example, the data is really interesting on male-female dyadic visits to nest boxes, but it would be useful to know a little more about this behaviour. For example for each focal male (or female), on average, with how many other individuals are such dyadic visits to nest boxes made? Does that differ between individuals that are subsequently faithful and unfaithful individuals? Are such dyadic visits made by individuals that are already in pairs? i.e. if a pair were in a social partnership the previous spring with the partnership persisting into the subsequent year.

Finally, it would also be useful if the authors are able to characterise the temporal patterns with respect to the dyadic nest box visits. Were they mostly in the late winter period closest to nesting, or did they occur in the early winter period.

I believe that the additional focus on the nest box visits is useful because this behaviour does appear to be more likely related to mate choice and pair settlement. The other form of data (social associations at feeders), is more likely to simply reflect the spatial distribution of birds and more incidental associations between neighbouring birds (although I take the point that this has been accounted for in the analyses presented).

Is it possible for the authors to make it a little clearer, the extent to which their main results can be driven by a) the spatial proximity of individuals, and the social associations. It seems clear enough that both are contributing to the incidence of extrapair paternity, but it would be useful if they are able to describe that clearly in percentage terms. Would be useful to make a clear statement about this in the discussion.

I was somewhat surprised at the relatively high number of individuals that were present in the winter, but not in the subsequent spring. Perhaps that is because in Europe, as opposed to populations in the UK, there is a lot of mobility across populations. Would be useful of the authors can discuss this a little more. For example is it the case that a lot of these birds have come from further north to over-winter in this area? Furthermore, with respect to the main findings, what would we expect in other populations such as those studied in the UK, where a higher proportion of the birds that are found in the winter will be present in the spring?

Referee: 2

Comments to the Author(s)

This observational study uses technology in combination with social network analysis as applied to an exceptional (possibly unique) data set to better understand the proximate determinants of social pair formation and, in particular, between-individual variation in extra-pair paternity

(EPP) in the socially monogamous blue tit. This excellent study addresses a generally very important topic which will be of major interest to readers of Proceedings B. The results are intriguing, the text is written exceptionally well and the authors do a particularly good job in communicating a relatively complex pipeline of statistical analyses and the corresponding results in a very concise manner. I have only one major point which the authors should consider discussing more (see below) and in addition a number of more specific comments which mainly point to opportunities for improving the presentation.

Major comments

To what degree do “winter associations” (as the main predictor of interest in this contribution) themselves represent carry-over effects from the preceding breeding season? For example, for non-yearling birds (as of start of the study in November), associations in winter may simply reflect social network structure from last spring’s breeding neighbourhood. In this context, the authors refer to an as yet unpublished study (reference 22) addressing effects of “changes in neighbourhood” (between breeding seasons) on within-individual variation in in extra-pair paternity. Lacking access to this piece of work, I am not sure whether the documented “little effect” can rule out a correlation/carry-over between associations during the preceding breeding season and associations in winter. Experimental manipulation of winter associations is probably very difficult if not impossible, but carry-over effects from the preceding breeding neighbourhood should not apply to yearling birds (as of start of the study in November). It could thus be interesting to explore whether effects of winter association on subsequent mating patterns differ by age category by including interaction terms (age category/dyad age combination-by-winter association). For example, if the reported significant main effect of winter association on EPP is mainly driven by dyads involving non-yearling birds (as of start of the study in November), this would be suggestive of carry-over effects from the preceding breeding neighbourhood to winter time.

Specific comments (lines)

- 54 and 91: socially monogamous OR socially-monogamous (but ideally consistently).
- The correct statement in lines 60-61 would be supported more convincingly if also null results were featured in the preceding sentence (lines 57 to 60).
- 167: Delete redundant “initial”.
- 209: Please add some information on the observed variation in minimum time differences here (e.g. SD and range).
- 292: Delete redundant “in winter”.
- 327: Please use wording that more clearly indicates whether 89% refers to pairs or individuals.
- 334: Give N.
- 471/ Figure 1: For the analysis visualised in Figure 1, neighbourhood order is the dependent and winter association strength is the independent variable (as also explicitly stated in line 476 nearby). Therefore I do not understand why neighbourhood order is on the x and winter association strength is on the y in Figure 1?
- 482-483/ legend of Figure 2: Figure 2 visualizes model predictions including uncertainty and thus the legend should state this and inform about the model underlying the predictions. For example, are the presented predictions controlled for other significant effects in the corresponding model (e.g. age)?
- 493/ Figure 3: For x axis labeling it seems preferable to use exactly the same terms as established and used in the text, e.g. “2nd order neighbours” instead of “2nd direct neighbours” (also applies to supplement); the abbreviation “EP” can be easily guessed but is nowhere defined.
- 519-522/ Table 1: The text states a positive relationship but both estimates are negative. It will less likely cause confusion for readers trying to understand the table content without reference to the text, when in line 520 the measure of breeding proximity (neighbourhood order) is given. Also give N.
- 537: Replace “direct” with “1st”.
- 538: Replace “neighbours” with “neighbourhoods”.

Author's Response to Decision Letter for (RSPB-2019-2606.R0)

See Appendix A.

Decision letter (RSPB-2019-2606.R1)

27-Jan-2020

Dear Miss Beck

I am pleased to inform you that your manuscript entitled "Winter associations predict social and extra-pair mating patterns in a wild songbird" has been accepted for publication in Proceedings B.

Open Access

Paper charges

Sincerely,

Dr Daniel Costa
Editor, Proceedings B
mailto: proceedingsb@royalsociety.org

Associate Editor:

Comments to Author:

The authors have addressed all comments raised by the referees. I am totally happy with the manuscript in its current version.

Appendix A

Response to Referees

We thank the editors and the reviewers for taking the time to review our manuscript. The comments we received were very constructive, and addressing them has significantly improved our manuscript. We are submitting a revised version of our paper for your consideration and we believe that we addressed all concerns thoroughly and revised the paper accordingly.

Major changes in our revised manuscript:

Reviewer 1 requested to add more information on the nestbox visit patterns.

- We now provide more information on (a) the number of individuals with whom a focal individual visited, (b) on whether these numbers differed between faithful and unfaithful birds and (c) on the temporal changes in nestbox visits over the course of winter (presented in an additional Figure).

Reviewer 2 requested to examine whether winter associations simply carry over from the preceding breeding social structure

- To address this question we split our main analysis on extra-pair mating patterns in two: one including data of dyads with individuals that were present in the previous breeding season and one including dyads without previous experience. In both cases, the winter association strength predicted patterns of extra-pair paternity. Thus, we conclude that winter associations and future extra-pair mating patterns cannot solely result from the preceding breeding social structure.

- We revised Figure 1. The figure now shows the distribution of the winter association strength for each of the five neighbourhood orders (more consistent with the design of Figure 3). In addition, we added an analysis comparing the mean winter association strength between all possible neighbourhood order compositions similar to the analysis comparing different relationship types (see section: *“Comparing the association strength between social partners, extra-pair partners and other close neighbours”*).

The above changes strengthen our findings and none of our changes contradict previous results. Our detailed responses to the reviewer comments are given below.

Associate Editor

Board Member: 1

Comments to Author:

Great paper and a wonderful and important contribution to the field. Is there a reason why male age, but not female age is used in the analysis? Would it be possible to add that?

Thank you. We added only male age as this has been shown to influence extra-pair siring success in several species (e.g. Cleasby, I. R., & Nakagawa, S. (2012), Ibis), including in blue tits (results from our

study population: Schlicht, L. et al. (2015) *Journal of Animal Ecology*). We did not include female age, because it has no effect on extra-pair paternity (our unpublished data).

Reviewer(s)' Comments to Author:

Referee: 1

Comments to the Author(s)

This is an excellent study that provides probably the most detailed analysis of spatial and social relationships between breeding birds within a population, ever undertaken. This framework, achieved through an intensively tagged and monitored (with pit tags and decoders on feeders and nest boxes), provides an excellent opportunity to then investigate the effects of social and spatial structuring on patterns of infidelity in this socially monogamous bird. The question is an interesting one, given that despite many decades of study we still don't fully understand the variation in extrapair paternity across the individuals in a population.

The introduction and methods are both very well written and I have no general or specific comments on those sections.

Whilst there is no problem with the data and results presented in the study, with respect to the winter associations between male-female dyads and both social pairs and extrapair relationships it would be really good if the authors are able to provide a bit more information to help characterise these dyadic interactions. For example, the data is really interesting on male-female dyadic visits to nest boxes, but it would be useful to know a little more about this behaviour. For example for each focal male (or female), on average, with how many other individuals are such dyadic visits to nest boxes made? Does that differ between individuals that are subsequently faithful and unfaithful individuals? Are such dyadic visits made by individuals that are already in pairs? i.e. if a pair were in a social partnership the previous spring with the partnership persisting into the subsequent year. Finally, it would also be useful if the authors are able to characterise the temporal patterns with respect to the dyadic nest box visits. Were they mostly in the late winter period closest to nesting, or did they occur in the early winter period.

I believe that the additional focus on the nest box visits is useful because this behaviour does appear to be more likely related to mate choice and pair settlement. The other form of data (social associations at feeders), is more likely to simply reflect the spatial distribution of birds and more incidental associations between neighbouring birds (although I take the point that this has been accounted for in the analyses presented).

We thank the reviewer for all the constructive comments. In our revised manuscript, we have provided more information in the Results section on the number of individuals with whom a focal individual performed a visit and whether these differed between faithful and unfaithful birds. We further elaborate on the temporal changes in nestbox visits and added an additional figure in the supplementary material (lines 399-405 (line numbers refer to the manuscript in the appendix including tracked changes), Figure S1). There were no previous social partners from 2017 that ended up as extra-pair partners in 2018, and when quantifying the nestbox visits we excluded social pairs. Thus, we could not examine differences in the patterns of nestbox visits between newly and re-mated pairs. Investigating the temporal pair formation process of social and extra-pair partners is interesting, but is a study in-and-of itself.

Is it possible for the authors to make it a little clearer, the extent to which their main results can be driven by a) the spatial proximity of individuals, and the social associations. It seems clear enough that both are contributing to the incidence of extrapair paternity, but it would be useful if they are

able to describe that clearly in percentage terms. Would be useful to make a clear statement about this in the discussion.

In our revised manuscript we have now centred and standardized our explanatory variables (see lines 281-282, 315-317). This allows us to compare the effect sizes of spatial proximity and winter association strength on the likelihood of a male-female dyad to gain extra-pair young. We now also elaborate on the difference in effect sizes in the discussion (lines 476-479).

I was somewhat surprised at the relatively high number of individuals that were present in the winter, but not in the subsequent spring. Perhaps that is because in Europe, as opposed to populations in the UK, there is a lot of mobility across populations. Would be useful of the authors can discuss this a little more. For example is it the case that a lot of these birds have come from further north to over-winter in this area? Furthermore, with respect to the main findings, what would we expect in other populations such as those studied in the UK, where a higher proportion of the birds that are found in the winter will be present in the spring?

Blue tits are partial migrants (e.g. Winkel, W. & Frantzen, M. (1991), *Journal für Ornithologie*) but unfortunately we do not have information about where the blue tits that were present in our study site in winter come from, i.e. whether any of them were migrants from northern populations. Many of the individuals present in winter may also breed in the forest surrounding the study area. How differences in winter social structure affect mating behaviour is a very interesting question. In populations where many individuals only stay a limited period of time and will leave the area before breeding, birds will have a larger number of unique social associates (i.e. more individuals with which they have foraged together). How their association strength to “key” individuals (e.g. future social and extra-pair partners) is affected by this is unknown—we could predict that it will be stronger (i.e. those that remain resident bond together), that it will be weakened (i.e. individuals keep having to switch social partners as those that leave are lost), or it could be unchanged (i.e. if the identity of the partner dictates the strength of the association). In more stable populations, one could imagine that being exposed to more potential extra-pair partners during the winter (i.e. more individuals that will be present during the breeding season) will promote more extra-pair matings because individuals have a greater choice of potential partners. In the revised discussion we briefly mention that studying populations with different social structures would be very interesting and of value to better understand (extra-pair) mating behaviour (lines 508-510).

Referee: 2

Comments to the Author(s)

This observational study uses technology in combination with social network analysis as applied to an exceptional (possibly unique) data set to better understand the proximate determinants of social pair formation and, in particular, between-individual variation in extra-pair paternity (EPP) in the socially monogamous blue tit. This excellent study addresses a generally very important topic which will be of major interest to readers of *Proceedings B*. The results are intriguing, the text is written exceptionally well and the authors do a particularly good job in communicating a relatively complex pipeline of statistical analyses and the corresponding results in a very concise manner. I have only one major point which the authors should consider discussing more (see below) and in addition a number of more specific comments which mainly point to opportunities for improving the presentation.

Major comments

To what degree do “winter associations” (as the main predictor of interest in this contribution) themselves represent carry-over effects from the preceding breeding season? For example, for non-

yearling birds (as of start of the study in November), associations in winter may simply reflect social network structure from last spring's breeding neighbourhood. In this context, the authors refer to an as yet unpublished study (reference 22) addressing effects of "changes in neighbourhood" (between breeding seasons) on within-individual variation in extra-pair paternity. Lacking access to this piece of work, I am not sure whether the documented "little effect" can rule out a correlation/carry-over between associations during the preceding breeding season and associations in winter. Experimental manipulation of winter associations is probably very difficult if not impossible, but carry-over effects from the preceding breeding neighbourhood should not apply to yearling birds (as of start of the study in November). It could thus be interesting to explore whether effects of winter association on subsequent mating patterns differ by age category by including interaction terms (age category/dyad age combination-by-winter association). For example, if the reported significant main effect of winter association on EPP is mainly driven by dyads involving non-yearling birds (as of start of the study in November), this would be suggestive of carry-over effects from the preceding breeding neighbourhood to winter time.

We thank the reviewer for this very interesting comment. To address this question, we now split our main analysis on extra-pair mating patterns in two: one dataset only including dyads where both partners had been breeding in our study site in 2017 (here the preceding social structure could carry-over into winter association strength) and one dataset including dyads where at least one bird was unfamiliar to the site (here the winter association strength could not have resulted from carry-over effects). In both cases, the winter association strength predicted patterns of extra-pair paternity (lines 323-329, 411-414 (line numbers refer to the manuscript in the appendix including tracked changes), Table S3). Thus, winter associations and future mating patterns cannot solely result from the preceding breeding social structure. In our discussion, we note that this makes logical sense, as all individuals will always experience a flocking period during winter prior to their first breeding event (lines 483-490). However, it will be interesting for future studies to examine in more detail to which extent the preceding breeding structure can carry-over into the winter social structure and how this may change with different yearly carry-over rates.

Specific comments (lines)

-54 and 91: socially monogamous OR socially-monogamous (but ideally consistently).

Changed in line 54.

-The correct statement in lines 60-61 would be supported more convincingly if also null results were featured in the preceding sentence (lines 57 to 60).

We added additional references showing null results in lines 59-60.

-167: Delete redundant "initial".

Deleted.

-209: Please add some information on the observed variation in minimum time differences here (e.g. SD and range).

We added information on the SD, median and range in line 210.

-292: Delete redundant "in winter".

Deleted.

-327: Please use wording that more clearly indicates whether 89% refers to pairs or individuals.

Changed in line 370.

-334: Give N.

Added in line 377, 379.

-471/Figure 1: For the analysis visualised in Figure 1, neighbourhood order is the dependent and winter association strength is the independent variable (as also explicitly stated in line 476 nearby). Therefore I do not understand why neighbourhood order is on the x and winter association strength is on the y in Figure 1?

Thank you for pointing this out. Indeed, y and x axis were reversed. We changed this accordingly and decided to structure Figure 1 in the same way as Figure 3. Figure 1 now shows the distribution of the winter association strength for each of the five neighbourhood orders. In addition, we added an analysis comparing mean winter association strength between all possible neighbourhood order compositions (lines 296-302) similar to the analysis comparing different relationship types (see section: *“Comparing the association strength between social partners, extra-pair partners and other close neighbours”*).

-482-483/legend of Figure 2: Figure 2 visualizes model predictions including uncertainty and thus the legend should state this and inform about the model underlying the predictions. For example, are the presented predictions controlled for other significant effects in the corresponding model (e.g. age)?

Figure 2 did not control for other predictors. We changed this in the revised Figure 2 and provide more information about the underlying model.

-493/Figure 3: For x axis labeling it seems preferable to use exactly the same terms as established and used in the text, e.g. “2nd order neighbours” instead of “2nd direct neighbours” (also applies to supplement); the abbreviation “EP” can be easily guessed but is nowhere defined.

We changed this in Figure 3 and the supplementary material, as suggested.

-519-522/Table 1: The text states a positive relationship but both estimates are negative. It will less likely cause confusion for readers trying to understand the table content without reference to the text, when in line 520 the measure of breeding proximity (neighbourhood order) is given. Also give N.

We added more information about the measure of breeding proximity and hope that it is better understandable. See text of Table 1.

-537: Replace “direct” with “1st”.

Changed in text of Table 2.

-538: Replace “neighbours” with “neighbourhoods”.

Changed in Table 2.